# Whole Exome Sequencing of 20 Spanish Families: Candidate Genes for Non-Syndromic Pediatric Cataracts

**DOI:** 10.3390/ijms241411429

**Published:** 2023-07-13

**Authors:** Patricia Rodríguez-Solana, Natalia Arruti, María Nieves-Moreno, Rocío Mena, Carmen Rodríguez-Jiménez, Marta Guerrero-Carretero, Juan Carlos Acal, Joana Blasco, Jesús M. Peralta, Ángela Del Pozo, Victoria E. F. Montaño, Lucía De Dios-Blázquez, Celia Fernández-Alcalde, Carmen González-Atienza, Eloísa Sánchez-Cazorla, María de Los Ángeles Gómez-Cano, Luna Delgado-Mora, Susana Noval, Elena Vallespín

**Affiliations:** 1Molecular Ophthalmology Section, Institute of Medical and Molecular Genetics (INGEMM), IdiPaz, La Paz University Hospital, 28046 Madrid, Spain; prsolana@salud.madrid.org (P.R.-S.); mariarocio.mena@salud.madrid.org (R.M.); crodriguezj@salud.madrid.org (C.R.-J.); victoriaeugeniafdezmontano@hotmail.com (V.E.F.M.); carmenglezatienza@gmail.com (C.G.-A.); eloisasancaz@gmail.com (E.S.-C.); 2Department of Pediatric Ophthalmology, IdiPaz, La Paz University Hospital, 28046 Madrid, Spain; natalia.arruti@salud.madrid.org (N.A.); maria.nieves@salud.madrid.org (M.N.-M.); martu_gue@hotmail.com (M.G.-C.); juancarlos.acal@salud.madrid.org (J.C.A.); yoana.blasco@yahoo.es (J.B.); jesuspc22@hotmail.com (J.M.P.); celia.far93@gmail.com (C.F.-A.); susana.noval@salud.madrid.org (S.N.); 3European Reference Network on Eye Diseases (ERN-EYE), La Paz University Hospital, 28046 Madrid, Spain; 4Biomedical Research Center in the Rare Diseases Network (CIBERER), Carlos II Health Institute (ISCIII), 28029 Madrid, Spain; ingemm.adelpozo@gmail.com (Á.D.P.); mariadelosangeles.gomez@salud.madrid.org (M.d.L.Á.G.-C.); lunadelde@gmail.com (L.D.-M.); 5Clinical Bioinformatics Section, Institute of Medical and Molecular Genetics (INGEMM), IdiPaz, CIBERER, La Paz University Hospital, 28046 Madrid, Spain; ingemm.ldedios@gmail.com; 6Clinical Genetics Section, Institute of Medical and Molecular Genetics (INGEMM), IdiPaz, CIBERER, La Paz University Hospital, 28046 Madrid, Spain

**Keywords:** non-syndromic pediatric cataracts, bilateral cataracts, lenses, whole exome sequencing, ophthalmogenetics

## Abstract

Non-syndromic pediatric cataracts are defined as opacification of the crystalline lens that occurs during the first years of life without affecting other organs. Given that this disease is one of the most frequent causes of reversible blindness in childhood, the main objective of this study was to propose new responsible gene candidates that would allow a more targeted genetic approach and expand our genetic knowledge about the disease. We present a whole exome sequencing (WES) study of 20 Spanish families with non-syndromic pediatric cataracts and a previous negative result on an ophthalmology next-generation sequencing panel. After ophthalmological evaluation and collection of peripheral blood samples from these families, WES was performed. We were able to reach a genetic diagnosis in 10% of the families analyzed and found genes that could cause pediatric cataracts in 35% of the cohort. Of the variants found, 18.2% were classified as pathogenic, 9% as likely pathogenic, and 72.8% as variants of uncertain significance. However, we did not find conclusive results in 55% of the families studied, which suggests further studies are needed. The results of this WES study allow us to propose *LONP1*, *ACACA*, *TRPM1*, *CLIC5*, *HSPE1*, *ODF1*, *PIKFYVE*, and *CHMP4A* as potential candidates to further investigate for their role in pediatric cataracts, and *AQP5* and locus 2q37 as causal genes.

## 1. Introduction

### 1.1. Cataract Formation: Lens Structure and Function

Cataracts are defined as opacification of the crystalline lens and are differentiated according to whether they appear during the first years of life (congenital and infantile) or progressively with age (“age-related”) [1].

The lens is a transparent structure that, through a process of accommodation, focuses and refracts light onto the retina. There, photoreceptors detect it and convert it into visual signals that are transmitted by the optic nerve to various parts of the brain. The lens is derived from the superficial ectoderm and underlying mesenchyme and belongs to the so-called anterior tissues of the eye. Together with the cornea, it forms part of the refractive system [2].

Structurally, the lens consists of the capsule, the epithelium, the fibrous cells, the transition zone at the lens equator, and the cortex (Figure 1) [2].

The capsule has a protective function against external pathogens and facilitates the differentiation of lens cells and their adhesion to the epithelium [2]. The epithelium consists of progenitor cells that divide continuously throughout life. The transition zone is composed of epithelial progenitor cells, which, after mitosis, begin their differentiation into fibrous cells. As they differentiate, these fibers produce a considerable amount of lens protein, called crystallin, and degrade their nuclei and organelles by ubiquitination or autophagy [3]. During the process of differentiation, the fibers migrate toward the center of the lens, overlapping; thus, the nucleus of the lens is made up of old differentiated fibers, whereas the more undifferentiated fibers are located in the periphery, leading to the cortex [4].

**Figure 1 ijms-24-11429-f001:**
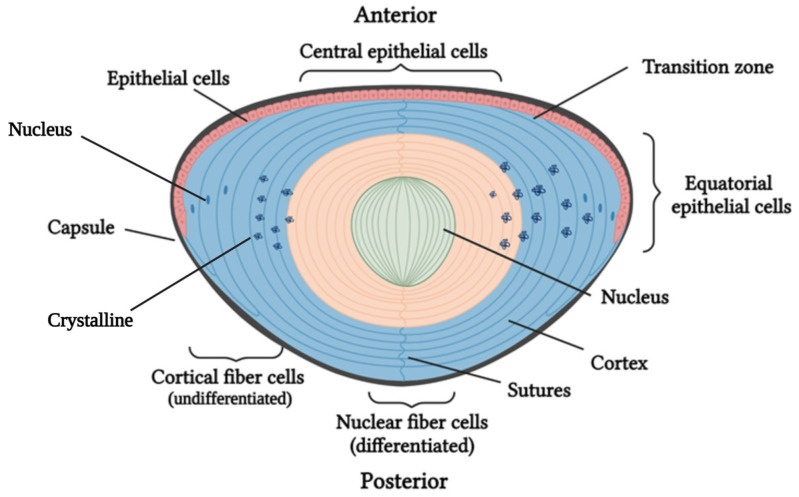
Diagram of the structural formation of the lens, which consists of the capsule, epithelial cells, and undifferentiated fibrous cells at the periphery, forming the cortex, and mature fibrous cells toward the interior, forming the nucleus [4]. Created by BioRender [5].

The correct functioning of the lens depends on its transparency. This transparency is achieved by the precise organization of its cells, the compact packing of proteins, and the correct degradation of nuclei and organelles. Alterations in the stability or structure of the lens can lead to light scattering, causing decreased vision [6].

### 1.2. Classification of Pediatric Cataracts

Pediatric cataracts refer to opacification of the lens that occurs during the first years of life. Several classification systems have been developed based on onset, morphology, and laterality.

Based on the onset, pediatric cataracts can be differentiated into congenital cataracts (defined as any lens opacity present at birth or within the first year of life) and infantile cataracts (diagnosed in older babies or children). The diagnosis of lens opacity at a later age does not exclude a congenital onset, unless the red reflex was properly checked at birth [7].

Regarding morphology, the most widely used classification system was first proposed by Merin et al., in 1971 [8]. In addition to morphology, knowing the location of the opacity is essential for its characterization and classification [9].

According to laterality, cataracts are called bilateral if both eyes are affected or unilateral if only one eye is involved.

Bilateral congenital cataracts are generally more common than unilateral congenital cataracts, which are mostly caused by local abnormalities, such as persistent foveal vasculature [7,10].

### 1.3. Pediatric Cataracts: Prevalence and Etiology

Pediatric cataracts are among the most common causes of preventable severe childhood visual impairment or blindness, with an incidence of 1.8–3.6/10,000 per year. The prevalence ranges from 0.32 to 22.9/10,000 children [7]. Underdeveloped and low-income areas have a higher prevalence, probably due to socioeconomic conditions and a higher risk of infectious diseases, such as rubella or herpes simplex virus, which are strongly associated with cataracts.

Furthermore, pediatric cataracts have a varying etiology, with the majority being idiopathic (62%); the remaining 38% being hereditary (25%) or non-hereditary (13%), associated with metabolic pathologies, trauma, or secondary cataracts associated with syndromic diseases [10].

Hereditary cataracts can be divided into syndromic (if they are part of a systemic disease involving other organs) and non-syndromic (in which the eye is the only organ affected).

### 1.4. Mechanism of Pediatric Cataract Development

It has been shown that α-crystallin interacts only with partially denatured proteins that tend to aggregate [11]; thus, most denatured or partially denatured crystallins are expected to be bound to α-crystallin. However, if the mutation produced is severe, some crystallins might not be bound and solubilized by α-crystallin [12]. Furthermore, if the amount of denatured crystallins or the speed of denaturation exceeds the capacity of the α-crystallin to act, this could lead to the presence of heavy aggregates that scatter light and damage the systems responsible for the homeostasis of the lens cells, compromising their survival.

Regarding the structural organization of the lens, the destruction of its microarchitecture observed in cataracts is the result of cell damage and degeneration caused by mutant crystallins [13]. Evidence from several studies [12,13] suggests that the damage observed is the result of a direct toxic effect of the mutant protein on lens cells, either by escaping the action of the chaperone (α-crystallin) or by saturating it.

One way in which crystallins can directly damage lens cells is through the induction of an unfolded protein response (UPR), followed by apoptosis. The UPR is a set of adaptive intracellular signaling pathways that reduces the stress produced in the endoplasmic reticulum as a consequence of misfolded protein accumulation. Under conditions of homeostasis, heat shock protein 5 (*HSPA5*) binds to at least three endoplasmic reticulum sensors (IRE1, ATF6, and EIF2AK3), keeping them in an inactive state. In the presence of large amounts of unfolded protein, *HSPA5* dissociates from these three sensors, activating them and initiating the UPR. When activated, this response results in increased chaperone synthesis, reduced transcription, and increased protein degradation through upregulation of endoplasmic reticulum-associated degradation protein levels. If these measures do not restore cellular integrity, the apoptotic pathway is initiated [12]. The lens histology observed in many types of hereditary cataracts could be due to the toxicity generated from mediators released by the UPR and apoptosis. Indeed, several studies in the literature support this theory [14,15,16].

### 1.5. Genetics of Pediatric Cataracts

Congenital cataracts were the first disease with autosomal dominant inheritance mapped in humans [17]. Although this pattern of inheritance predominates in pediatric cataracts of hereditary origin, autosomal recessive [18] and X-linked [19] patterns of inheritance have also been reported.

Non-syndromic congenital cataracts typically have complete penetrance, with autosomal dominant cataracts more common than recessive cataracts. Their expressivity is highly variable; thus, the same mutation within an affected family can give rise to various phenotypes. Furthermore, it is possible that the same type of cataract can be caused by mutations at different loci, making it a polygenic disease [6]. This complicated genotype–phenotype correlation complicates the genetic diagnosis of patients.

Currently, some 71 loci mapped to 56 cataract-causing genes have been identified [20]. This information is recorded in Cat-Map, a database that presents a chromosomal map together with the spectrum of mutations associated with hereditary cataracts, updated in real time [21]. The genetic causes of Mendelian cataracts can be grouped according to the cellular processes affected, suggesting that these pathways are critical for the correct development or homeostasis of the lens. Thus far, the most frequently implicated genes are those for crystallins (33%), developmental or transcription factors (26%), connexins (18%), membrane proteins and transporters (11%), genes involved in lipid metabolism (8%), and intermediate filament and chaperone proteins (4%) [20].

#### 1.5.1. Crystallin Protein Genes

Crystallins comprise over 90% of the lens’ proteins and have the most fundamental place in the lens’ structure [22]. They are responsible for maintaining the transparency and structural organization of the lens. There are three types of crystallins: α, β, and γ [23]. The α-crystallins are a multimeric protein complex composed of two oligomers, αA-crystallins and αB-crystallins, which are encoded by two genes located on different chromosomes, *CRYAA* (21q22.3) and *CRYAB* (11q22-q22.3) [24]. The α-crystallins belong to the family of small heat shock proteins (sHsps). They are molecular chaperones that associate with misfolded proteins by helping facilitate their rearrangement or promote their degradation by other chaperones and cofactors [25].

Genetic mutations that cause decreased expression of functional αA-crystallins and αB-crystallins typically result in recessively inherited cataracts, such that their residual activity is sufficient to maintain lens transparency during childhood, even though this can result in increased susceptibility to environmental damage or age-related cataracts. An example is the p.Arg54Cys mutation in *CRYAA*, which generates total congenital cataracts in homozygosis; whereas, when it is present in heterozygosis, it results in small point opacities [26]. However, missense mutations, which result in a deleterious mutant protein, often cause autosomal dominant cataracts as a consequence of blocking the activity of the normal crystallin by a dominant negative mechanism or by causing damage to the homeostasis of the lens.

On the other hand, β-crystallins can be made up of various numbers of monomers, resulting in dimers to octamers, whereas γ-crystallins are monomers [24]. In contrast to α-crystallins, most mutations that occur in these proteins produce an unstable protein that denatures immediately or under stress conditions, precipitating in the cytosol, scattering light, and leading to cataracts [27].

#### 1.5.2. Membrane Protein Genes

The elongation undergone by epithelial cells in their transition to fibrous cells requires the synthesis of large amounts of membrane proteins and lipids. Thus, any alteration that interferes with the synthesis of these structural components can lead to cataracts.

In relation to lipid synthesis, there are several studies that associate inherited cataracts with mutations in enzymes involved in the synthesis of various types of lipids, such as acylglycerol kinase [28] or lanosterol synthase [29].

In reference to membrane proteins, ion or solute transporters and proteins that maintain gap junctions are directly related to the appearance of bilateral congenital cataracts. Mutations in the *SLC16A12* gene, which affects an active transmembrane transporter of monocarboxylic acids, can cause juvenile cataracts with dominant inheritance [30]. Mutations in *DNMBP*, a guanine nucleotide exchange factor that regulates cell junction configurations, have also been described [31].

Connexins are a family of proteins consisting of hemichannels that mediate communication between the intra- and extracellular milieu and gap junctions that allow cell–cell communication between adjacent cells. The lack of lens vascularization means that cell nutrition and metabolite transport is performed through an extensive network of connexin channels. Mutations affecting some of these proteins can lead to cataractogenesis [32]. An estimated 22% of familial cataracts are due to mutations in the *GJA3* and *GJA8* genes coding for connexin 46 and connexin 50, respectively [33].

#### 1.5.3. Cytoskeletal Protein Genes

Beaded fiber-specific proteins belong to the family of intermediate filament proteins and form specific structures of the lens. The *BFSP1* gene and *BFSP2* gene encode filaments whose functions are not yet clearly understood; however, they play an important role in maintaining lens transparency [34]. This role is illustrated by animal studies showing the presence of cataracts in *BFSP1* and *BFSP2* knock-out mice [35]. In addition, many cases of pediatric cataracts of dominant or recessive inheritance caused by mutations in *BFSP1* have been reported in the scientific literature [36,37]. In fact, pediatric cataracts are the phenotype associated with *BFSP1* and *BFSP2* in the Online Mendelian Inheritance in Man (OMIM) database [38,39].

#### 1.5.4. Transcription Factors and Lens Growth Factors

One of the transcription factors most recently associated with congenital cataracts is paired box gene 6 (*PAX6*). This transcription factor is involved in the development of lenses and central nervous system tissues. Mutations affecting the C-terminal domain have been associated with cataracts and corneal dystrophy [40].

*EPHA2* also plays an essential role in the development of lens and central system tissues. It is part of the ephrin receptor tyrosine kinase family of receptors and its mutations can lead to dominant or recessive early onset cataracts or even increase the risk of age-related cataracts [6,41,42].

#### 1.5.5. Degradation Proteins or Chaperones

As previously mentioned, lens formation requires extensive restructuring of its components and differentiation of epithelial cells into fibrous cells. Fundamental to this process is the removal of cellular organelles and high levels of protein degradation. Mutations that disrupt this degradation process can lead to cataracts. An example is mutations in *FYCO1*, a scaffolding protein active in the microtubule transport of autophagic vesicles, which cause autosomal recessive cataracts [43], suggesting that autophagic vesicles are important for organelle degradation in developing lens fiber cells, among other functions. Another example involves *CHMP4B*, which encodes a member of the charged multivesicular body/chromatin-modifying protein (CHMP) family of proteins. It is a component of the major subunits of the endosomal sorting complex required for transport III (ESCRT-III) machinery, which facilitates innumerable biological processes of membrane remodeling and scission [44]. One of these processes is autophagy, which is a mechanism of cellular self-degradation essential for the removal of aggregated proteins and damaged organelles. Thus, impaired autophagolysosomal degradation could explain cataract formation in the absence of proper CHMP4B function [45]. It has been reported that mutations in *CHMP4B* can cause polar and subcapsular cataracts [46].

#### 1.5.6. Metabolism-Associated Genes

The robust metabolism of lens cells maintains cellular homeostasis, thus providing a reducing environment to cope with the oxidative damage to which lens cells are subjected, preventing osmotic stresses and inhibiting glycosylation products.

Several genes involved in metabolism have been reported to cause cataracts over time. *GALK1* encodes galactokinase, which is a major enzyme for the metabolism of galactose. Its deficiency causes pediatric cataracts during infancy and presenile cataracts in the adult population [47]. *ALDH1A1* encodes the next enzyme after alcohol dehydrogenase in the major pathway of alcohol metabolism. Studies on mice have shown that ALDH1A1 protects the eye against cataract formation via an enzymatic pathway by detoxifying particulate matter generated by oxidative stress [48].

### 1.6. Symptoms and Treatment of Pediatric Cataracts

As previously discussed, late diagnosis of congenital cataracts can result in their mislabeling as infantile cataracts. Early diagnosis of congenital cataracts and prompt surgical intervention are essential to preventing irreversible amblyopia. If retinal blurring occurs during the critical period of visual development, vision could be lost. Therefore, therapeutic measures during the period of sensory plasticity are of vital importance to prevent blindness.

The most common clinical sign of congenital cataract is leukocoria, which is the white coloration of the pupil when examined under a beam of light. Other common signs are nystagmus (defined as involuntary, rapid, and repetitive movement of the eyes) and strabismus (the loss of parallel alignment of the eyes) [49]. In addition to these manifestations, non-syndromic congenital cataracts can occur in association with other ocular defects that compromise eye function, such as aniridia (absence of iris), microcornea (horizontal diameter less than 11 mm), or microphthalmia (small eye size) [50,51].

Currently, the only treatment for congenital cataracts is surgery. It has been shown that increasing age at the time of surgery correlates with poorer visual outcomes. The optimal time period for cataract removal is 8–10 weeks. Lambert and colleagues showed in their study that 88% of children operated on before 10 weeks of age achieved a score of 20/80 or better, compared with those operated on after 10 weeks, with a score of 20/100 or poorer [52]. Depending on the age of the patients and the anatomical characteristics of the eye, an intraocular lens (IOL) may be implanted after lensectomy, a process called pseudophakia. If only a lensectomy is performed, the eye is left aphakic and refractive correction with contact lenses or thick glasses is necessary [53].

Some studies have shown that visual outcomes in patients with bilateral congenital cataracts appear to be better after primary IOL implantation compared with aphakia corrected with contact lenses [54], although the rate of re-operations in the IOL-implanted children was significantly higher than in the left aphakic group [55,56,57].

The identification of the precise genetic mutation facilitates accurate genetic counseling, providing families with information and ongoing support for their condition, which is crucial for these patients. It also provides a more accurate prognosis, which is particularly important in syndromic cases [58].

All this demonstrates the importance of early detection of pediatric cataracts; it is therefore essential to expand our genetic knowledge of the disease. Therefore, the aim of this study is to propose new candidate genes responsible for non-syndromic pediatric cataracts, which will serve as a reference for other researchers and will allow us to offer better health care to patients with cataracts at the National Health System’s La Paz University Hospital, given that it is a reference center for the diagnosis and treatment of this pathology (Centers, Services and Reference Units, CSUR).

## 2. Results

### 2.1. Ophthalmological Examination

Of the 23 patients included in our study, 45 eyes were analyzed (1 patient had a unilateral cataract; this patient was included in our study, as was the sibling of another patient with bilateral cataracts). The results are summarized in Table 1.

Lensectomy was performed in 28 eyes. Eighteen eyes remained unoperated by the end of the study period because the cataracts were visually insignificant or visual acuity remained above driving-standard levels. Surgery was performed before the age of 1 year in 16 (59%) eyes. Postoperative glaucoma was found in six eyes. Eight eyes had nystagmus at the end of the study period and ten eyes had microphthalmia (Figure 2).

The most common type of cataracts in our cohort were nuclear and lamellar, present in 12 eyes in both types (Figure 3).

### 2.2. Molecular Genetics

After performing a whole exome sequencing (WES) analysis, we were able to identify the causal gene for pediatric cataracts in 10% (2/20) of the cohort and thus concluded the genetic diagnosis. In 35% (7/20) of the cohort, genes were found which, due to their relationship with the crystalline lens or with the cataract formation process, we considered to possibly be candidate genes responsible for the phenotype, although it would be necessary to perform further studies to confirm this. As for the remaining 55% (11/20) of the families, the result of the WES was inconclusive (Figure 4). Of the 10% of families with a genetic diagnosis and 35% with candidate genes, we found 11 variants. Of these, 18.2% (2/11) were classified as pathogenic (P), 9% (1/11) as likely pathogenic (LP), and 72.8% (8/11) as a variant of uncertain significance (VUS) (Table 2, Figure 5). As for pathogenic variants, a variant in *AQP5* (NM_001651.3: c.152T>C:p.(Leu51Pro)) and a deletion at the 2q37.3 locus have been previously described, demonstrating that this specific variant causes congenital cataracts.

## 3. Discussion

As previously mentioned, pediatric cataracts have a highly variable expression. Together with the polygenicity and heterogeneity described, this variability makes it difficult to describe new genes associated with this phenotype. In addition, given that the exome occupies less than 2% of the entire human genome, it is possible that the WES study would not be able to identify the cause of the pathology. All this could explain why 55% of the families analyzed by WES attained inconclusive results, necessitating further studies on the variants found. Moreover, at present, the cost-effectiveness ratio of WES versus whole genome sequencing shows that it would be interesting to implement the latter in all the patients who remain without a genetic diagnosis to date [64] and, whenever possible, to start using this technique as the first diagnostic method.

However, the study of the whole exome remains an alternative solution in some cases. In 10% of the families analyzed, a genetic diagnosis was achieved. A de novo variant was found in the OFT-00040 family proband in the *AQP5* gene, NM_001651.3:c.152T>C:p.(Leu51Pro), which had been previously reported in a study from Qingdao of posterior subcapsular congenital cataracts [63], demonstrating the causality of the variant. In the other diagnosed family, OFT-00350, a de novo deletion was found in the 661.2 Kb proband in the region 2q37.3. This region had been associated with the occurrence of bilateral posterior polar cataracts in two Chinese families [60]. This phenotype also coincided with the phenotype of our patient, which showed that the deleted region was clearly associated with pediatric cataracts.

In 35% of the cohort, genes were found that could be candidates for causing pediatric cataracts due to their relationship with the pathologic processes of the disease. In some cases, the evidence was stronger than in others, as in the *CHMP4A* and *PIKFYVE* genes. However, we considered it important to report the findings to serve as a precedent for other researchers.

This result highlights the current need to search for new candidate genes that will help us delimit the genetic etiology of pediatric cataracts, thus contributing to improving patient care. The results obtained are specifically discussed below.

The proband in the family OFT-00247 was heterozygous for the mutation in the *LONP1* gene, (OMIM 605490), NM_004793.3:c.1939G>A:p.(Glu647Lys), inherited from his father, who did not have lens opacity.

*LONP1* encodes a mitochondrial matrix protein that belongs to the Lon family of ATP-dependent proteases. This protein mediates the selective degradation of misfolded, unassembled, or oxidatively damaged polypeptides in the mitochondrial matrix. It could also have a chaperone function [65]. Diseases associated with *LONP1* include cerebral, ocular, dental, auricular, and skeletal anomalies syndrome, which has pediatric cataracts as an associated phenotype [66]. In Patel, Nisha et al.’s study [67], the *LONP1* gene was the second most commonly mutated gene in the pediatric cataract cohort after the crystallin genes, affecting five families. Our variant had previously been reported in homozygosity in two siblings with congenital cataracts and mild dysmorphia [59]. Although our patient was heterozygous for the variant and did not present mild dysmorphia, mutations in *LONP1* associated with non-syndromic pediatric cataracts present in heterozygosis have been described in the literature [68]. Moreover, variable expressivity of the phenotype in individuals from the same family described in the literature could explain the paternal inheritance of the variant without lens opacities in the father [68]. It was also possible that, taking into account what has been described about the NM_004793.3:c.1939G>A:p.(Glu647Lys) variant, there was a second event in the proband that we were not able to find through the whole exome study.

The missense variant found [59] causes the substitution of a negatively charged amino acid and glutamic acid (E) for a positively charged amino acid and a lysine (K), at position 647. Given that the positions that remain conserved in a protein throughout evolution are relevant to its function and that the Pdel value (probability of a deleterious effect of the variant as calculated by the Panther Classification System) is 0.85, the variant could have a possibly harmful effect on the protein function [69]. This hypothesis is reinforced by studying possible changes in protein structure by employing tools such as Polymorphism Phenotyping v2 (PolyPhen-2) [70].

The importance of protein degradation in the eye as explained above suggests the pathogenic nature of the variant; however, further functional studies would be necessary to conclusively determine its pathogenic nature. Therefore, this variant is classified as of uncertain significance by the ACMG criteria [71].

In the OFT-00289 family, both siblings were heterozygous for the variant in the *ACACA* gene (OMIM 200350), NM_198839.2:c.1126C>T:p.(Arg376Cys), inherited from their healthy mother through germline mosaicism. Acetyl-CoA carboxylase alpha (*ACACA*) encodes one form of the complex multifunctional enzyme system acetyl-CoA carboxylase (ACC). The isoform encoded by *ACACA* maintains the regulation of fatty acid synthesis, producing malonyl-CoA, which is the first intermediary in cytosolic fatty acid synthesis and is a key metabolite for long-chain saturated fatty acid production.

Although the *ACACA* gene has been ruled out as a causal gene for pediatric cataracts in a previous study [72], many studies have shown the metabolic relevance of this enzyme. Activity-impeded ACACA reduces cytoplasmic membrane fluidity and impairs mobilities of transmembrane receptors, ultimately impairing cell membrane-dependent biological processes [73]. Moreover, inhibition of ACC can increase the intracellular acetyl-CoA level and stimulate the influx of calcium into the cells [74], which in the lens have been demonstrated to possibly be responsible for carboxypeptidase E activation and formation of a truncated form of αB-crystallin in cataracts [75].

The missense variant found had extremely low frequency in gnomAD population databases, affecting a gene with a gnomAD Z-score greater than 3.09 (7.24) and for which the predictive tools suggested a deleterious effect of the gene. In addition, this variant was the only one that both affected siblings did not share with their parents. However, due to insufficient evidence to date, it was classified as a VUS.

In reference to family OFT-00334, the female proband presented bilateral cataracts and was heterozygous for the mutation in the *TRPM1* (OMIM 603576) gene, NM_001252024.2:c.4720dup:p.(Ser1574LysfsTer7), inherited from her mother, who did not have cataracts. It was possible that there was a second mutation that we were not able to find through exome sequencing.

Although *TRPM1* encodes a calcium-permeable cation channel, which has been associated with autosomal recessive congenital stationary night blindness (CSNB), a case has recently been described in a 4-week-old boy of non-consanguineous parents affected by unilateral nuclear cataracts [76]. Most cases of unilateral pediatric cataracts are presumed to be sporadic; however, some families with unilateral cataracts have been reported, suggesting in some cases a germline mutation as an underlying cause [77].

Saleh E. et al. reported two novel presumed pathogenic variants that were identified in blood and lens epithelial cells in *TRPM1*. Following the genetic diagnosis, full-field electroretinography showed ON-bipolar signaling defects in both rods and cones compatible with CSNB [76], a phenotype that was not initially suspected. Moreover, homozygous animal models for *TRPM1* have also been described, showing incipient posterior cortical cataracts and immature cataracts [78]. In addition, an important paralog of *TRPM1* is *TRPM3*, with approximately 57% amino acid sequence identity. Heterozygous mutations in this gene have been linked to early onset cataracts with or without other eye abnormalities [79].

Regarding the genetic study of our proband, the frameshift variant found generated a premature stop codon that produced a truncated protein in a gene where loss of function is a known mechanism of disease. This characteristic, coupled with the extremely low frequency in gnomAD population databases, classified the variant as likely pathogenic by the ACMG criteria. Although our patient had bilateral cataracts and not unilateral cataracts, we considered it of interest to report as a candidate gene for future functional studies related to pediatric cataracts.

The male proband of the OFT-00350 family, who presented bilateral posterior polar cataracts, was heterozygous for a de novo deletion of 661.2 Kb, located in locus 2q37.3, which included 17 genes classified as pathogenic by the ACMG guidelines for copy number variants (CNVs) [80]. The deletion was found by a CNV study obtained from exome sequencing, confirmed by a single nucleotide polymorphism (SNP) array in the proband and excluded in the parents.

Currently, 71 cataract-associated loci have been mapped [20], among which the 2q37-qter locus was described in 2012 [60]. This article describes two Chinese families in which all affected members had a clear diagnosis of non-syndromic bilateral posterior polar congenital cataracts that were present after birth and developed during childhood. Genetic scans based on dinucleotide repeat microsatellite markers allowed the authors to describe a chromosomal region near the terminal end of chromosome 2q implicated in cataract development. This region comprised more than 40 known genes, of which they classified *CXCR7*, *AQP12B*, and *SEPT2* as candidate genes based on their position and functionality.

The deleted region in our patient shared only one of the genes proposed as a candidate by Ouyang, the gene for aquaporin 12B (*AQP12B*). However, it also included the *AQP12A* gene, which was not specified in Ouyang’s article. Both genes encode transmembrane transporters of water and small neutral solutes.

Given that the lens is an avascular non-innervated organ, a hydrostatic pressure gradient generated by the activity of pumps, channels, and other transport proteins directs the supply of nutrients and the removal of metabolic waste from the lens. This mechanism substitutes for blood flow and maintains the ocular properties of the lens involved in transparency and light refraction [81]. Various studies have shown the development of early onset bilateral cataracts in *AQP0* and *AQP1* knockout mice [82,83]. Because of its importance as described in the lens, it is possible that the *AQP12B* or *AQP12A* present in the deletion loci in our patient were playing an essential role in the phenotype.

Although its pathogenicity should be verified with additional studies, the 661.2 kb deletion has been classified as pathogenic by the ACMG guidelines [80], suggesting its involvement in the development of pediatric cataracts.

In family OFT-00338, both siblings were heterozygous for the variant NM_016929.4:c.514C>T:p.(Arg172Trp) in the *CLIC5* gene (OMIM 607293), which was inherited from their affected mother, who also presented the variant in heterozygosity.

This gene encodes a member of the chloride intracellular channel (CLIC) family of chloride ion channels. Although mutations in the human *CLIC5* gene have been linked with progressive autosomal recessive non-syndromic sensorineural hearing impairment with or without vestibular dysfunction, *CLIC5* has recently been localized to cilia and/or centrosomes in the lens, describing an unknown essential role in normal fiber cell organization and formation of the lens suture in the eye [84]. Given that the sutures form along the optical axis of the lens, their precise organization is important for light transmission.

As for the missense variant found, it involves the replacement of arginine, which is basic and polar, by tryptophan, which is neutral and slightly polar, at codon 172 of the CLIC5 protein. Its population frequency in gnomAD is extremely low; however, algorithms developed to predict the effect of missense changes on protein structure and function are either unavailable or do not agree on the potential impact of this missense change (SIFT: “Deleterious”; PolyPhen-2: “Possibly Damaging”; Align-GVGD: “Class C0”) [70,85,86]. It has been classified as a VUS. Therefore, despite the insufficient and contradictory information available on this variant, recent evidence on the *CLIC5* gene in the lens supports proposing it as a possible causal gene for the OFT-00338 family phenotype.

In reference to family OFT-00346, the female proband was heterozygous for the frameshift variant NM_002157.2:c.61_62insACCA:p.(Ser21AsnfsTer5) in the *HSPE1* gene (OMIM 600141) and for the deletion NM_024410.4:c.678_686del:p.(Cys227_Pro229del) in *ODF1* (OMIM 182878). Both undescribed variants appeared de novo in the patient, whose parents were healthy.

The *HSPE1* gene encodes an important heat shock protein, which functions as a chaperonin, preventing protein misfolding and promoting the refolding and proper assembly of unfolded polypeptides generated under stress conditions in the mitochondrial matrix. Together with Hsp60, it facilitates the correct folding of imported proteins [87].

The *ODF1* gene is considered by sequence conservation to be a member of the sHsps family of proteins, of which crystallins and the *HSPE1* gene mentioned above also form a part [88]. For this reason, the *ODF1* gene is alternatively called heat shock protein beta-10 (*HSPB10*).

The sHsps belong to molecular chaperones, which protect prokaryotic and eukaryotic cells against the deleterious effects of stress, preventing stress-induced irreversible aggregation of damaged proteins and facilitating renaturation of bound substrates cooperating with other molecular chaperones [89]. Several articles demonstrate a protective role of sHsps against eye diseases such as cataracts and glaucoma [90]. Therefore, the sHsps family is implicated in cataract formation, because aggregation of misfolded proteins is one of the most common pathogenic mechanisms in cataracts.

The frameshift variant found in the *HSPE1* gene has not been previously described and its population frequency in gnomAD is extremely low (0.0%). Moreover, a premature stop codon is generated that can cause loss of normal protein function by truncation or nonsense-mediated mRNA decay, resulting in an absent or altered protein product. However, given that there is thus far insufficient information about the variant, it has been classified as a VUS.

The deletion found in the *ODF1* gene has an extremely low frequency in the gnomAD population databases. Also, the resulting protein is shorter in length due to the in-frame deletion in a non-repeat region, which affects the functionality of the protein. Based on ACMG criteria, the variant has been classified as a VUS.

Lastly, although functional models are needed to prove the pathogenicity of the variants, the important role played by the *HSPE1* and *ODF1* genes in the lens and the de novo presence of the variants in the proband led us to propose both genes as causal for pediatric cataracts.

The OFT gene panel study previously performed by our group in family OFT-00215 [91] revealed that the male proband was heterozygous for the variant NM_014290.2:c.1085C>T:p.(Pro362Leu) in the *TDRD7* gene. Given that it had autosomal-recessive inheritance and the second mutation was not found, the study was extended to WES, which showed that he was also heterozygous for NM_015040.3:c.5844+3A>G in the *PIKFYVE* gene (OMIM 604632), inherited from his unaffected mother.

This gene encodes a lipid kinase that phosphorylates phosphatidylinositol in phosphoinositide lipid formation, which is a signaling molecule that regulates critical cell processes. PIKFYVE is the only source of phosphatidylinositol 3,5-bisphosphate (PI(3,5)P2) whose levels dynamically and transiently change in response to specific stimuli. In mammalian cells, physiological signals such as insulin and growth factors cause an acute elevation of its levels, suggesting that this lipid plays key roles in cellular homeostasis and in adaptation [92]. In addition, many studies have indicated that PIKFYVE plays important roles in lysosomes, causing the appearance of enlarged vacuoles [93], as well as in ionic homeostasis and membrane dynamics [94] and in transcription regulation [92].

Although *PIKFYVE* is associated with corneal dystrophy, disruption of *PIKFYVE* has recently been reported to cause pediatric cataracts in humans and zebrafish [95]. In this article, a possible pathogenic variant, NM_015040.4:c.5828G>A:p.(Gly1943Glu), was found in *PIKFYVE* by WES in a Chinese family with congenital cataracts and without corneal defects. The mutation was located in the phosphatidylinositol phosphate kinase (PIPK) domain of the PIKFYVE protein. Functional studies of the variant in animal models have confirmed that the *PIKFYVE* gene might also be involved in the development of early onset cataracts in zebrafish [95].

Regarding the genetic study of our proband, the splicing variant found (NM_015040.3:c.5844+3A>G) was located in an intronic region of the PIPK domain, which could affect splicing by affecting the essential kinase activity of the enzyme, as shown in recent functional studies [95]. This variant has not been previously described and it has an extremely low frequency in the gnomAD population database. However, computational prediction tools unanimously support a benign effect on the gene for a splice region variant. Therefore, due to contradictory information regarding this variant, it has been classified as a VUS by the ACMG guidelines [71].

In family OFT-00235, the male proband had a biallelic mutation in *CHMP4A* (OMIM 610051). He presented the nonsense variant, NM_014169.5:c.406C>T:p.(Gln136Ter), inherited from his unaffected father, and the frameshift variant, NM_014169.5:c.65del:p.(Ala22GlufsTer5), inherited from his healthy mother.

CHMP4A belongs to the chromatin-modifying protein/charged multivesicular body protein (CHMP) family. This protein comprises part of ESCRT-III, a complex involved in the formation of multivesicular bodies, which are delivered to lysosomes, allowing the degradation of membrane proteins such as stimulated growth factor receptors, lysosomal enzymes, and lipids [96]. In addition, the GeneCards database [97] associates *CHMP4A* with cataract 32 [98], which includes early onset lamellar cataracts, the phenotype presented by our proband. An important paralog of this gene is *CHMP4B*, which has been described as necessary for lens growth and differentiation in animal models [99] and as responsible for autosomal dominant infantile cataracts [100].

The biallelic mutation has been classified as a VUS because current information on these variants is thus far contradictory. However, knowing the importance of protein and metabolic degradation in the lens, it would be expected that an alteration in the *CHMP4A* pathway would lead to lens opacities, thus classifying this gene as a possible causative gene for pediatric cataracts. Even so, functional studies of the variants would be necessary to determine their pathogenic effect.

For family OFT-00040, the female proband was heterozygous for a de novo missense variant in the aquaporin 5 (*AQP5*) (OMIM 600442) gene, NM_001651.3:c.152T>C:p.(Leu51Pro). *AQP5* encodes for a small transmembrane water channel protein, which plays a redundant role in water transport in the eye, lung, and sweat glands [101].

As discussed for other members of the aquaporin family (*AQP12B*, *AQP0*, *AQP1*) in the OFT-00350 family, water flow is essential for the maintenance of transparency and light refraction in the lens. In addition to these aquaporins, *AQP5* has been reported to be expressed in epithelial cells and in lens fiber cells [102,103] and its expression changes as a function of fiber cell differentiation from an intracellular localization in differentiating fiber cells to the plasma membrane of mature fiber cells due to the loss of fiber cell nuclei. This characterization would indicate that AQP5 also plays a key role in the maintenance of lens properties. Recently, mutations in *AQP5* have been directly related to the development of autosomal dominant congenital cataracts [63].

The missense variant found in our proband has been previously described in a family from Qingdao with posterior subcapsular congenital cataracts [63]. Tang, Suzhen et al. observed that the lens of *AQP5*-KO mice showed mild opacity at approximately 6 months of age, thus demonstrating that the mutation NM_001651.3:c.152T>C:p.(Leu51Pro) in *AQP5* was associated with autosomal dominant congenital cataracts. Therefore, the variant has been classified as pathogenic, taking into account the following ACMG criteria: its extremely low frequency in the gnomAD database; computational prediction tools unanimously supporting a deleterious effect for a missense variant; de novo appearance in a patient with a consistent phenotype; existence of well-established functional studies that show the damaging effect on the gene or gene product. For these reasons, we can conclude that the mutation NM_001651.3:c.152T>C:p.(Leu51Pro) in the *AQP5* gene was the causal gene for pediatric cataracts in the proband.

## 4. Materials and Methods

A combined ophthalmological and genetic study was performed by the Multidisciplinary Unit of Ophthalmogenetics at La Paz University Hospital, in accordance with the principles of the Declaration of Helsinki, and it was approved by the hospital’s ethics committee.

The inclusion criteria were the following: (1) Spanish pediatric patients with cataracts; (2) absence of syndromic phenotype; (3) negative result in sequencing study implementing a panel (OFTv2.1) that included 39 known non-syndromic pediatric cataract disease genes. Whenever possible, the patients’ relatives (parents, aunts, uncles, or siblings) were included in the study. In this way, 23 affected children from 20 unrelated families were recruited. After signing the informed consent, WES was performed.

### 4.1. Ophthalmological Studies

The demographic data collected during this study were eye laterality, cataract type, microphthalmia, nystagmus, age at surgery, and development of postoperative glaucoma.

### 4.2. Genetic Studies

Second, a genetic analysis was performed using genomic DNA obtained from leukocytes isolated from a peripheral blood sample. This was performed in the pre-analytical area of our institute, employing the Chemagic Magnetic Separation Module I (Chemagen, PerkinElmer, Waltham, MA, USA). DNA concentrations were measured by quantification with a NanoDrop ND-1000 spectrophotometer (Thermo Fisher Scientific, Waltham, MA, USA). Library preparation was performed with the Nextera DNA Exome Illumina DNA Prep with Enrichment (Illumina, San Diego, CA, USA) and IDT for Illumina DNA/RNA UD Indexes Set, A, B, C, or D, Tagmentation. Sequencing was performed on the high-quality sequencers, HiSeq4000 and NovaSeq6000, capturing 19,433 genes with the xGen™ Exome Research Panel v2 IDT.

The first analysis was then performed by the Bioinformatics of Medical and Molecular Genetics (INGEMM) team, which developed an analytical algorithm that identified SNPs and CNVs (insertions and deletions of small DNA fragments (indels)) within the capture regions of the exome.

The system comprised a sample pre-processing step, alignment of reads to a reference genome, identification and functional annotation of variants, and variant filtering. All these steps employed open tools widely used in the scientific community, as well as proprietary tools. In addition, all steps were robustly designed and included parameters that allowed the process to be monitored and the appropriate quality controls to deliver a reliable report on the variants in question.

The bioinformatics analysis was performed by the Clinical Bioinformatics Unit of the INGEMM center, employing the following software tools: trimmomatic-0.36, bowtie2-align version 2.0.6, picard-tools 1.141, samtools version 1.3.1, bedtools v2.26, and GenomeAnalysisTK version 3.3-0. The databases used were dbNSFP version 3.5, dbSNP v151, ClinVar date 20180930, ExAC-1, SIFT ensembl 66, Polyphen-2 v2.2.2, MutationAssessor release 3, FATHMM v2.3, CADD v1.4, and dbscSNV1.1.

CNV analysis was conducted utilizing in-house software called LACONv, specifically designed to calculate the copy number dosage of individual exons in targeted gene panels with the appropriate adjustment for GC sequence content. The copy number dosage of each captured region in the test sample was compared against all other samples in the same panel run to control for the batch effect. The P-values were computed using Mann–Whitney U tests; CNVs were deemed statistically significant if the value exceeded 0.05.

The second data analysis consisted of assessing the clinical pathogenic significance of the variants found in the patients by accessing multiple databases. These included PubMed [104], GeneCards [97], Decipher [105], gnomAD [106], UCSC Genome Browser [107], and specific databases such as Cat-Map, which is an online reference chromosomal map for inherited and age-related cataracts in humans, mice, and other vertebrates [108]. A literature search was performed for the gene and the variant found. The variants were then classified according to American College of Medical Genetics and Genomics (ACMG) prioritization [71,80] criteria as pathogenic, probably pathogenic, a VUS, probably benign, or benign, using the Varsome and Franklin database [109,110].

In some cases, candidate variants were included in GeneMatcher [111], an open-access website that allows connections to be established between researchers around the world who share an interest in the same gene. Depending on the type of variant, validation was performed by Sanger sequencing or by other techniques, such as SNP arrays.

## 5. Conclusions

In conclusion, WES was performed on 20 families with pediatric cataracts, reaching a diagnosis rate of 10%, in which two known mutations were found in the *AQP5* gene (OMIM 600442) and in the 2q37 locus.

Furthermore, WES allowed us to propose several candidate genes that could cause pediatric cataracts due to their relationship with the lens and its transparency in 35% of the families. These genes are *LONP1* (OMIM 605490), *ACACA* (OMIM 200350), *TRPM1* (OMIM 603576), *CLIC5* (OMIM 607293), *HSPE1* (OMIM 600141), *ODF1* (OMIM 182878), *PIKFYVE* (OMIM 604632), and *CHMP4A* (OMIM 610051).

In these cases, it will be necessary to continue with studies at the genomic, functional, familiar, and population levels to determine whether these genes are causative of pediatric cataracts and whether the combination of mutations in several genes could be causing them, following an oligogenic inheritance.

It is important to share the data obtained in public databases as well as to establish collaborations (e.g., ERN-EYE) to add knowledge and to be able to make new hypotheses about the genetic causes of pediatric cataracts, especially in undiagnosed families. In our study, we were not able to find variants that we suspected might be causing the pediatric cataract phenotype in 55% of the families.

Furthermore, we would consider extending the study in undiagnosed families, using new techniques such as long-read sequencing, whole genome sequencing, transcriptome analysis, or DNA methylation, which would allow us to reach a genetic diagnosis that would facilitate accurate genetic counseling.

## Figures and Tables

**Figure 2 ijms-24-11429-f002:**
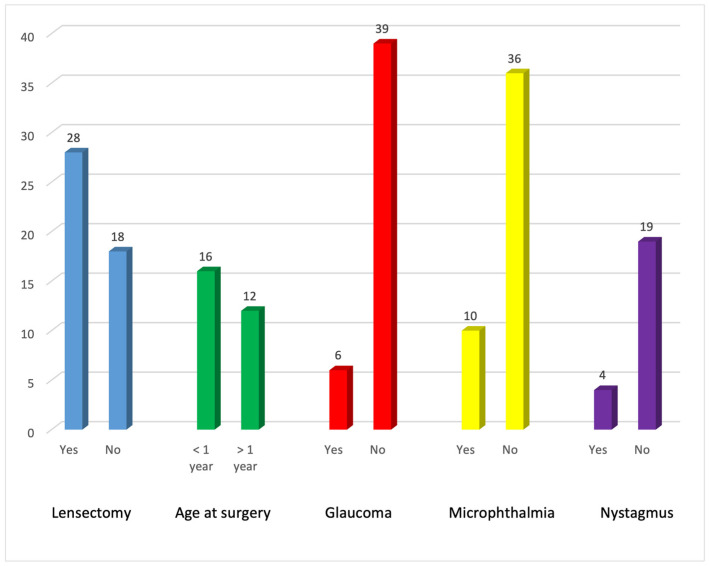
Representation of lensectomies performed in our cohort, age at surgery, and postoperative glaucoma, nystagmus, and microphthalmia.

**Figure 3 ijms-24-11429-f003:**
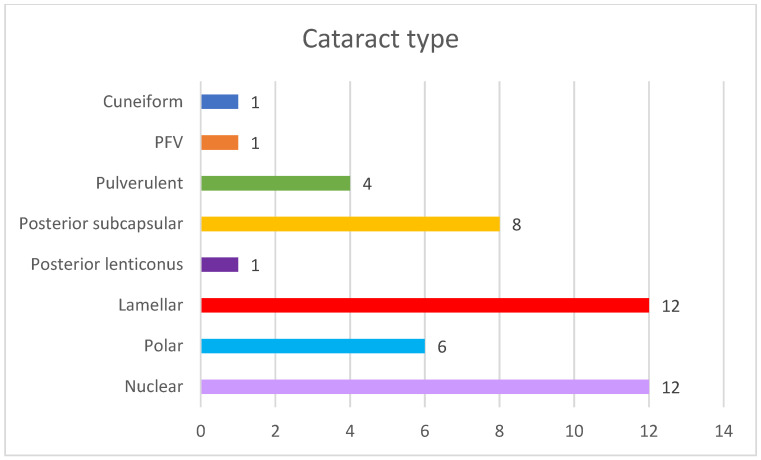
Cataract types in our cohort.

**Figure 4 ijms-24-11429-f004:**
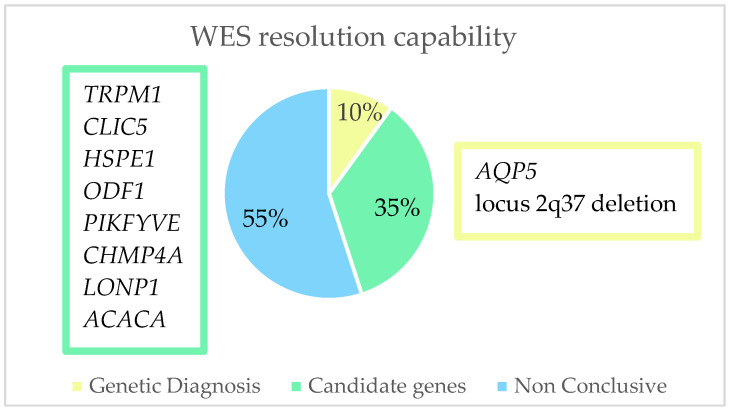
Representation of the resolution capacity of whole exome sequencing in this study. We were able to find the causal genes (*AQP5* and deletion in locus 2q37) in 10% of the cohort; in 35% of the families, we found genes that could be responsible for the formation of pediatric cataracts; in 55% of the cohort we did not find any conclusive results.

**Figure 5 ijms-24-11429-f005:**
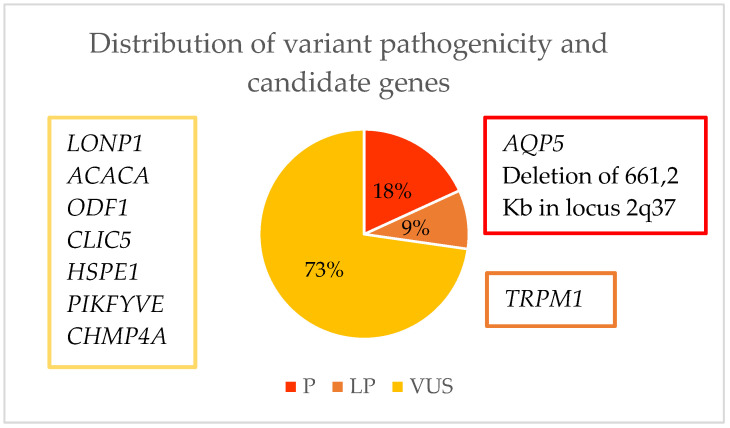
Representation of the possible candidate genes for causing pediatric cataracts and the pathogenicity of the variants found.

**Table 1 ijms-24-11429-t001:** Ophthalmologic findings.

Family ID	Relationship	Eye	Cataract Type	Microphthalmia	Nystagmus	Age at Surgery	Glaucoma
OFT-00172	Proband	RE	Nuclear	Yes	No	<1 year	Yes
		LE	Nuclear	Yes	No	<1 year	No
OFT-00247	Proband	RE	Posterior polar	Yes	No	N/A	N/A
		LE	PFV	Yes	No	<1 year	Yes
OFT-00289	Proband	RE	Posterior subcapsular	No	No	>1 year	No
		LE	Posterior subcapsular	No	No	>1 year	No
	Sister	RE	Lamellar	No	No	>1 year	No
		LE	Lamellar	No	No	>1 year	No
OFT-00222	Proband	RE	Nuclear	No	Yes	<1 year	Yes
		LE	Cuneiform	No	Yes	N/A	N/A
OFT-00334	Proband	RE	Posterior subcapsular	No	No	N/A	N/A
		LE	Posterior polar	No	No	N/A	N/A
OFT-00350	Proband	RE	Posterior polar	No	No	>1 year	No
		LE	Posterior polar	No	No	N/A	N/A
OFT-00214	Proband	RE	Nuclear	Yes	Yes	<1 year	Yes
		LE	Nuclear	Yes	Yes	<1 year	No
OFT-00322	Proband	RE	Posterior polar	No	No	N/A	N/A
		LE	Posterior subcapsular	No	No	N/A	N/A
OFT-00338	Proband	RE	Posterior subcapsular	No	No	N/A	N/A
		LE	Posterior subcapsular	No	No	N/A	N/A
	Sister	RE	Transparent lens	No	No	N/A	N/A
		LE	Posterior lenticonus	No	No	>1 year	No
OFT-00346	Proband	RE	Lamellar	No	No	>1 year	No
		LE	Lamellar	No	No	>1 year	No
OFT-00302	Proband	RE	Lamellar	Yes	No	<1 year	No
		LE	Lamellar	Yes	No	<1 year	No
OFT-00215	Proband	RE	Lamellar	No	No	>1 year	No
		LE	Lamellar	No	No	>1 year	No
OFT-00235	Proband	RE	Lamellar	No	No	N/A	N/A
		LE	Lamellar	No	No	N/A	N/A
OFT-00377	Proband	RE	Pulverulent	No	No	N/A	N/A
		LE	Pulverulent	No	No	N/A	N/A
	Proband	RE	Pulverulent	No	No	N/A	N/A
		LE	Pulverulent	No	No	N/A	N/A
OFT-00040	Proband	RE	Nuclear	No	Yes	<1 year	No
		LE	Nuclear	No	Yes	<1 year	No
OFT-00446	Proband	RE	Nuclear	Yes	Yes	<1 year	Yes
		LE	Nuclear	Yes	Yes	<1 year	Yes
OFT-00345	Proband	RE	Lamellar	No	No	>1 year	No
		LE	Lamellar	No	No	>1 year	No
OFT-00456	Proband	RE	Nuclear	No	No	<1 year	No
		LE	Nuclear	No	No	<1 year	No
OFT-00487	Proband	RE	Posterior subcapsular	No	No	N/A	N/A
		LE	Posterior subcapsular	No	No	N/A	N/A
OFT-00522	Proband	RE	Nuclear	No	No	<1 year	No
		LE	Posterior polar	No	No	<1 year	No

Abbreviations: RE—right eye, LE—left eye, PFV—persistent foveal vasculature, N/A—not applicable because no surgery was performed.

**Table 2 ijms-24-11429-t002:** Genetic results of non-syndromic pediatric cataracts in a Spanish cohort.

Family ID	Sex	Relationship	Gene	Mutation	Location	ACMG Criteria *	ACMG Result	Coding Impact	Zygosity	Inheritance	De Novo/ Inherited	Reported By
OFT-00172	M	Proband	NC	NC	-	-	-	-	-	-	-	-
OFT-00247	F	Proband	*LONP1*	NM_004793.3:c.1939G>A p.(Glu647Lys)	Exon 13	PM2, PP3, BP6	VUS	Missense	Hetero	AD/AR	Paternal	(Ma, A., 2021) [59]
OFT-00289	M	Proband	*ACACA*	NM_198839.2:c.1126C>T p.(Arg376Cys)	Exon 15	PM2, PP2, PP3	VUS	Missense	Hetero	AD/AR	Maternal Germline Mosaic	Novel **
	F	Sister										
OFT-00222	M	Proband	NC	NC	-	-	-	-	-	-	-	-
OFT-00334	F	Proband	*TRPM1*	NM_001252024.2:c.4720dup p.(Ser1574LysfsTer7)	Exon 27	PVS1, PM2	LP	Frameshift	Hetero	AR	Maternal	Novel **
OFT-00350	M	Proband	Locus 2q37.3	Deletion of 661,2 Kb (chr2:241526318–242187541)	Chr 2	1A, 2A-2E, 2H, 3A, 5A	P	-	Hetero	AD/AR	De novo	(Ouyang, 2012) [60]
OFT-00214	F	Proband	NC	NC	-	-	-	-	-	-	-	-
OFT-00322	F	Proband	NC	NC	-	-	-	-	-	-	-	-
OFT-00338	M	Proband	*CLIC5*	NM_016929.4:c.514C>T p.(Arg172Trp)	Exon 5	PM2, BS2	VUS	Missense	Hetero	AR	Maternal	ClinVar [61]
	F	Sister										
OFT-00346	F	Proband	*HSPE1*	NM_002157.2:c.61_62insACCA: p.(Ser21AsnfsTer5)	Exon 2	PM2	VUS	Frameshift	Hetero	AD/AR	De Novo	Novel **
			*ODF1*	NM_024410.4:c.678_686del p.(Cys227_Pro229del)	Exon 2	PM2, PM4	VUS	Inframe Deletion	Hetero	AD/AR	De Novo	Novel **
OFT-00302	F	Proband	NC	NC	-	-	-	-	-	-	-	-
OFT-00215	M	Proband	*PIKFYVE*	NM_015040.3:c.5844+3A>G	Exon 39	PM2, PS3, BS4	VUS	Splicing	Hetero	AD	Maternal	Novel **
OFT-00235	M	Proband	*CHMP4A*	NM_014169.5:c.406C>T p.(Gln136Ter)	Exon 4	PM2, BS2	VUS	Nonsense	Hetero	AD/AR	Paternal	gnomAD [62]
				NM_014169.5:c.65del p.(Ala22GlufsTer5)	Exon 2	PM2	VUS	Frameshift	Hetero		Maternal	Novel **
OFT-00377	F	Proband	NC	NC	-	-	-	-	-	-	-	-
	F	Sister	NC	NC	-	-	-	-	-	-	-	-
OFT-00040	F	Proband	*AQP5*	NM_001651.3:c.152T>C p.(Leu51Pro)	Exon 1	PM2, PP3, PS2, PS3	P	Missense	Hetero	AD	De Novo	(Tang, 2021) [63]
OFT-00446	M	Proband	NC	NC	-	-	-	-	-	-	-	-
OFT-00345	F	Proband	NC	NC	-	-	-	-	-	-	-	-
OFT-00456	M	Proband	NC	NC	-	-	-	-	-	-	-	-
OFT-00487	F	Proband	NC	NC	-	-	-	-	-	-	-	-
OFT-00522	M	Proband	NC	NC	-	-	-	-	-	-	-	-

Abbreviations: F—female; M—male; NC—not conclusive; P—pathogenic; LP—likely pathogenic; VUS—variant of uncertain significance; Hetero—heterozygous; AD—autosomal dominant; AR—autosomal recessive; ACMG—American College of Medical Genetics and Genomics. * ACMG Criteria in Appendix A, Table A1 and Table A2; ** Not previously reported in the literature.

## Data Availability

Not applicable.

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
