# Peer review of "Whole Exome Sequencing of 20 Spanish Families: Candidate Genes for Non-Syndromic Pediatric Cataracts"

_ijms, 2023, doi:10.3390/ijms241411429_

Round 1

Reviewer 1 Report

This paper presented a whole exome sequencing (WES) study of 20 Spanish families with no syndromic pediatric cataracts and a previous negative result on an ophthalmology next-generation sequencing panel (NGS). It is a topic of interest to researchers and broadens the cataract mutation map, which is significant. But the novelty is still limited, and the style of this paper is more likely a review, not a research paper.

Major:

1.     This introduction part is too long, the author should restructure and refine the significance.

2.     The figure could be integrated into 1-2 figures. Several pie charts were suggested convert to bar charts.

3.     This paper is more likely a case report, more information should be included, such as the slit lamp picture, and pedigree diagram.

4.     The conclusion part also needs to be simplified and refined.

Quality of English language is good.

Author Response

Thank you very much for reviewing the manuscript. Below are the changes made following the reviewers' suggestions.

Reviewer 2 Report

Rodriguez-Solana and colleagues describe 20 families with pediatric non-syndromic cataracts and highlight possible candidate genes identified with WES analysis. The results are of potential interest but the description is confusing and many important data are lacking.

First, they state in discussion and abstract: “55% obtained inconclusive results”, but it seems that one single family out of 20 (OFT-00040) obtained conclusive results.

I think the authors should provide a more concise introduction, merge figure 2, 3 and 4 (they could easily be much smaller), thoroughly modify results and discussion.

The discussion should start from AQP5, which looks like the only pathogenic finding. Afterwards, I would discuss the 2q37.2 deletion and CHMP4 variants, which are very interesting candidates. Then, shortly, the other findings.

More specifically:

11)      How was the 2q37.2 deletion of patient OFT-350 identified??? It is possible to detect large heterozygous deletions from WES data, but nothing is stated in materials and methods. What algorithm was used? How was it confirmed in the proband and excluded in parents (array analysis, RT-PCR…)? The authors should provide an image regarding this result.

The authors claim this deletion overlaps a candidate region of congenital cataract, that was described in Chinese families through linkage analysis. Although AQP12B is a good candidate, this by no means proves that its haploinsufficiency causes congenital cataract. Other families with haploinsufficiency of this gene would be needed. Furthermore, the reference genome of the deletion is not reported, but it seems that AQP12A is involved as well. Why is this gene not considered as a good candidate? It is not correct to state (see table 3) that the deletion is reported in Ouyang et.: it is a different type of study and no variants were identified in AQP12B

22)      OFT-00040 carries the only causative variant, interesting because it confirms that this specific AQP5 variant causes congenital cataract. I would put this description on top of the  result section. AQP5 mutations have been associated to palmoplantar keratoderma (OMIM 600231); were any dermatological issues reported in this patient?

33)      OFT-00338 – CLIC5. Varsome evaluates this variant as likely benign. However, is the mother affected? If yes, what kind of cataract? Are there any other affected family members? It may be interesting to expand the segregation analysis.

44)      OFT-00247 – LONP1. Rather than a “candidate gene”, this looks like an inconclusive result: the very same variant was identified in homozygous state in 2 siblings with cataract and mild dysmorphism (Ma A. et al. Hum Mut 2020, “Genome sequencing in congenital cataract improves diagnostic yield”). I think that either this variant does not cause cataract in the family reported by Rodriguez-Solana, or there probably was a second variant in the gene, not detected by current methods.

55)      OFT-00289 – ACACA. ACACA mutations are a known cause of an autosomal recessive disorder (acetyl-CoA carboxylase deficiency), where there is no ocular involvement to my knowledge. Do authors think this novel variant may play a gain of function role? Furthermore, how could they state there is a MATERNAL germinal mosaicism?

66)      OFT00334 – TRPM1. Several “loss of function” variants in this gene are recessive alleles of congenital stationary night blindness… of course it is LP according to ACMG criteria: this patient and her mother are carriers of CSNB.

77)      OFT-00346 or OFT-350??? – HSPE1 and ODF1. Both de novo… Was paternity confirmed?? And maternity as well, this could be an adopted child. Anyway, evidence for causality of these variants is very weak.

88)      OFT-215 – PYKFYVE c.5844+3A>G is a good candidate, but it was inherited from an unaffected mother, whereas the pathogenic variant reported by Mei S. et al. is fully penetrant…

99)      OFT-235 – CHMP4A. This is a very good candidate! My suggestion to the authors is to put this finding in Gene Matcher.

Quality of Englis is not bad but it would need some editing

Author Response

(The authors gave the same response as above.)

Reviewer 3 Report

Authors conducted an interesting study to find  potential candidate genes in  Spanish families with non- syndromic pediatric cataracts using whole exome sequencing (WES). They could find a genetic diagnosis in 10% of the families studied and reported several possible causative genes involved in cataracts in 35% of their cohort.  However, they could not identify obvious findings in 55% of their families. Moreover, they proposed  LONP1, ACACA, TRPM1, CLIC5, 47 HSPE1, ODF1, PIKFYVE, and CHMP4A as potential candidates. 

However, it is recommended to consider following comments:

1- Regarding heterozygous mutations related to disease with AR pattern of inheritance, It is highly recommended to look for any reports of heterozygous mutation for these genes that could cause disease including cataracts. This could help strengthen their study with more evidence. Without any reports of heterozygous mutations involved in diseases, it cannot be possible to conclude the causative roles of them. To give an example, LONP1 is an ATP-dependent protease and chaperone that have multiple key roles in mitochondri and there is a report of a patient with heterozygous for a de novo mutation in LONP1: c.901C>T,p.R301W who presented seizures, encephalopathy, pachygyria and microcephaly. 

2- Some figures could be merged in one figure.

Author Response

(The authors gave the same response as above.)

Round 2

Reviewer 1 Report

The article is well improved, can be accepted.

Minor editing of English language required.

Reviewer 3 Report

Authors made substantial changes to their manuscript and can be acceptable.